# The Gender-Biased Differential Effect of *KDM6A* Mutation on Immune Therapy in Urothelial Carcinoma: A Public Database Study

**DOI:** 10.3390/cancers17030356

**Published:** 2025-01-22

**Authors:** Yohei Sekino, Hikaru Nakahara, Kenichiro Ikeda, Kohei Kobatake, Yuki Kohada, Ryo Tasaka, Kenshiro Takemoto, Shunsuke Miyamoto, Hiroyuki Kitano, Keisuke Goto, Akihiro Goriki, Keisuke Hieda, Nobuyuki Hinata

**Affiliations:** 1Department of Urology, Graduate School of Biomedical and Health Sciences, Hiroshima University, Hiroshima 734-8551, Japan; kenikeda@hiroshima-u.ac.jp (K.I.); kkobatake@hiroshima-u.ac.jp (K.K.); ykohada@hiroshima-u.ac.jp (Y.K.); tasakary@hiroshima-u.ac.jp (R.T.); take44@hiroshima-u.ac.jp (K.T.); sm0025@hiroshima-u.ac.jp (S.M.); tanokin@hiroshima-u.ac.jp (H.K.); keigoto@hiroshima-u.ac.jp (K.G.); agoriki1@hiroshima-u.ac.jp (A.G.); hiedak@hiroshima-u.ac.jp (K.H.); hinata@hiroshima-u.ac.jp (N.H.); 2Department of Clinical and Molecular Genetics, Graduate School of Biomedical and Health Sciences, Hiroshima University, Hiroshima 734-8551, Japan; hnkhr@hiroshima-u.ac.jp

**Keywords:** bladder cancer, KDM6A, mutation, immune therapy, gender bias

## Abstract

The genes that escape from X chromosome inactivation (XCI) contribute to gender differences in urothelial carcinoma (UC). *KDM6A* is one of the XCI genes. In this study, we identified a gender-biased difference in prognosis according to *KDM6A* mutation in the immune therapy cohort (IMvigor 210). The mutation rate of *KDM6A* was higher in female patients than in male patients in several cohorts in bladder cancer and upper tract urothelial carcinoma. Female patients with *KDM6A* mutation were associated with poorer prognosis than those in male patients in three immune therapy cohorts. Immune fraction analysis showed that female patients with *KDM6A* mutation had tumors with immune cold status compared to those in male patients. These results indicate that the effect of *KDM6A* mutation on immune therapy varied according to gender, and the status of *KDM6A* mutation may be a promising biomarker in immune therapy in UC.

## 1. Introduction

Urothelial carcinoma (UC), which includes both upper tract urothelial carcinoma (UTUC) and bladder cancer (BLCA), is one of the most common genitourinary malignancies worldwide. BLCA was the tenth most common cancer in 2020 [1]. BLCA can be classified into non-muscle-invasive bladder cancer (NMIBC) and muscle-invasive bladder cancer (MIBC). In NMIBC, T1 tumors constitute an aggressive subtype with 40% recurrence and 15% progression to MIBC at 5 years. MIBC will eventually develop distant metastasis, resulting in a 5-year survival rate of <50%. Although standard care for MIBC is neoadjuvant chemotherapy followed by radical cystectomy, about 40% of patients experience relapse [2]. UTUC is diagnosed as an invasive disease in 60% of cases and is associated with a poor prognosis [3]. Therefore, clarifying the molecular biology of the pathogenesis of UC is crucial for its clinical management.

The incidence rate of BLCA is three to five times higher in males than females [4], who are at higher risk of recurrence, progression, and cancer-specific mortality of BLCA than males [5]. Some studies reported a higher incidence in males than females in UTUC [6]. The high incidence of BLCA in males is said to be due to high smoking rates [7], but gender differences persist even after adjusting for risk factors such as cigarette smoking and occupational hazards in BLCA [8]. A recent study showed that high immune activity and basal-associated signatures were observed in female patients, whereas luminary papillary and neuroendocrine-like subtypes were enriched in male patients [9]. In advanced cancer, including BLCA treated with immune checkpoint inhibitors, progression-free survival (PFS) was longer in male patients than in female patients [10]. These findings suggest that genetic differences between genders may play a role in the survival disparities observed with immunotherapy in UC.

X chromosome inactivation (XCI) silences transcription from one of the two X chromosomes in females to balance gene amounts between females and males [11]. However, about 15% of X chromosome genes can escape the XCI process to protect females from complete functional loss of X chromosome genes by mutation [12]. A previous study found that among genes escaping from XCI, the loss of function mutation of six X-chromosome genes (*ATRX*, *CNKSR2*, *DDX3X*, *KDM5C*, *KDM6A*, and *MAGEC3*) occurred more frequently in males than females [12], indicating that these molecules may contribute to the gender-biased difference. Additionally, several studies have reported that some of these molecules were associated with tumor immune response [13,14,15,16]. Therefore, we hypothesized that these genes escaping from XCI might affect gender-biased immune responses in UC.

In this study, we analyzed the prognostic role of the above six X-chromosome genes using the IMvigor 210 clinical study and identified a gender-biased prognostic role in patients with *KDM6A* mutation. We then studied the comprehensive gender-biased role of *KDM6A* mutation in UC using several public databases.

## 2. Materials and Methods

### 2.1. Public Databases

The processed data from the American Association for Cancer Research (AACR) GENIE cohort were downloaded via AACR Project GENIE [17]. The processed data from the Cancer Genome Atlas (TCGA) Urothelial Bladder Carcinoma and International Cancer Genome Consortium (ICGC) were downloaded via the UCSC Xena tool [18]. The processed data from the study by Samstein et al. [19], the study by Clinton et al. [20], the UC-GENOME (Urothelial Cancer—Genomic Analysis to Improve Patient Outcomes and Research) study [21], and the study by Pietzak et al. [22] were downloaded via cBioPortal [23]. The processed data from UROMOL [24] were downloaded via source data. The processed data from the study by Taber et al. [25] were downloaded via Supplementary Tables. The processed data from the IMvigor 210 clinical study [26] were downloaded from the European Genome-phenome Archive (EGA; https://ega-archive.org/) and received approval from Roche (Boston, MA, USA). The processed data from the upper tract urothelial cancer (UTUC) study by Fujii et al. [27] were downloaded from the EGA (https://ega-archive.org/) and received approval from Roche. For mutation analysis, all mutation types (missense, truncating, splice, and other mutation types) were considered “mutated type”.

### 2.2. Immune Fraction Analysis

The processed data of the single-sample Gene Set Enrichment Analysis (ssGSEA) in the IMvigor 210 study were downloaded via Supplementary Tables [28]. The processed data of immune cell proportions by CIBERSORTx in the UC-GENOME study were downloaded via source data [23].

### 2.3. Statistical Analysis

Statistical differences were evaluated using the two-tailed Student *t*-test or Mann–Whitney U-test. Fisher’s exact test was used to evaluate gender-biased differences in a contingency table. After a Kaplan–Meier analysis, any statistical difference between the survival curves of the cohorts was determined with the log-rank Mantel–Cox test. A *p*-value of <0.05 was considered statistically significant. Statistical analyses were conducted primarily using GraphPad Prism software version 9 (GraphPad Software Inc., La Jolla, CA, USA) or JMP version 17 (SAS Institute, Cary, NC, USA). All results were visualized based on GraphPad (GraphPad Software Inc., La Jolla, CA, USA).

## 3. Results

### 3.1. Prognostic Role of Genes Escaping from X Chromosome Inactivation in Immune Therapy

We analyzed the gender-biased prognostic role of 6 XCI genes (*ATRX*, *CNKSR2*, *DDX3X*, *KDM5C*, *KDM6A*, and *MAGEC3*) in the immune therapy cohort of IMvigor 210 in which PD-L1 treatment with atezolizumab was tested in a phase 2 clinical trial in UC [26]. The mutations in *CNKSR2* and *MAGEC3* were not identified in the IMvigor 210 study. Among these XCI genes, a significant gender-biased difference in prognosis was observed only for the *KDM6A* gene. Kaplan–Meier analysis showed that male patients with *KDM6A* mutation were significantly associated with a favorable overall survival (OS), whereas no significant difference in OS was observed in the female patients in the IMvigor 210 study (Figure 1A–C, Appendix A). *KDM6A* is a gene that encodes a protein called lysine-specific demethylase 6A, which plays a crucial role in regulating gene expression by modifying histones [29]. The mutation of *KDM6A* is frequently observed in patients with UC [30]. Recent studies have reported that *KDM6A* showed more mutations in NMIBC in females than in males, and loss of *KDM6A* increases BLCA risks in female patients [31], indicating that *KDM6A* plays a decisive role in the gender difference in BLCA. However, the gender-biased role of *KDM6A* mutation in the immune response has not been fully elucidated. On the basis of these findings, we focused on the gender-biased role of *KDM6A* mutation in UC.

### 3.2. KDM6A Expression in Males and Females with Urothelial Cancer from Several Bladder Cancer and Upper Tract Urothelial Cancer Cohorts

As mentioned above, *KDM6A* is one of the genes that escape from XCI [12]. Therefore, we analyzed the mRNA expression of *KDM6A* in males and females in several UC cohorts, including BLCA and UTUC. The mRNA expression of *KDM6A* was higher in females than in males in several of the examined cohorts (Figure 2A–F).

### 3.3. Rate of KDM6A Mutations in Males and Females from Several Bladder Cancer and Upper Tract Urothelial Cancer Cohort

A recent study reported that the *KDM6A* mutation rate was higher in females than males in NMIBC [31]. The UROMOL study [24], as well as the studies by Pietzak et al. [22] and Clinton et al. [20] have provided us with genomic information on NMIBC. The *KDM6A* mutation rate was higher in females than in males in the UROMOL study and the study by Clinton et al. (Table 1). Then, we studied the gender difference in the *KDM6A* mutation rate in several BLCA and UTUC cohorts. The rate of *KDM6A* mutation was higher in the females than the males in the AACR GENIE cohort of BLCA (*p* = 0.014), UTUC cohort (*p* = 0.019), and the UTUC study by Fujii (*p* = 0.031). There were no significant differences in *KDM6A* mutation rates in the other cohorts (Table 2 and Table 3).

### 3.4. Gender-Biased Prognostic Role of KDM6A Mutation in Non-Muscle Invasive Bladder Cancer

The UROMOL study provided prognostic findings (recurrence-free survival (RFS) and PFS) in NMIBC. The Kaplan–Meier curve showed that the female patients with the *KDM6A* mutation were associated with a poor RFS and male patients were associated with a favorable PFS in the UROMOL study (Figure 3A–F). These results indicate that female patients with *KDM6A* mutation may be associated with a poorer prognosis than male patients in NMIBC.

### 3.5. Gender-Biased Prognostic Role of KDM6A Mutation in Bladder Cancer

A recent study showed that low expression and mutation of *KDM6A* were associated with poor prognosis in female patients with BLCA in the TCGA cohort [32]. Then, we performed the gender-biased prognostic role of *KDM6A* mutation at several endpoints (PFS/OS) from TCGA cohorts. The Kaplan–Meier curve showed that the female patients with *KDM6A* mutation were associated with poor PFS and OS (Figure 4A–F). Additionally, we studied the gender-biased prognostic role of *KDM6A* mutation in the AACR GENIE cohort [17], the ICGC cohort [33], and the study by Clinton et al. [20]. The male patients with *KDM6A* mutation were significantly associated with alive status, but there was no significant prognostic difference between wild-type and mutated-type *KDM6A* in the male patients in the AACR GENIE cohort (Table 4). The Kaplan–Meier curve showed that the female patients with *KDM6A* mutation were associated with poor OS, but there was no significant prognostic difference between wild-type and mutated-type *KDM6A* in the male patients in the ICGC cohort (Figure 4G–I). The Kaplan–Meier curve showed that the male patients with *KDM6A* mutation were associated with a favorable OS, but there was no significant prognostic difference between wild type and mutated type in the female patients in the study by Clinton et al. (Figure 4J–L). These results indicate that the prognostic impact of *KDM6A* mutations varies by gender. Female patients with *KDM6A* mutation may be associated with a poorer prognosis than male patients.

### 3.6. Gender-Biased Prognostic Role of KDM6A Mutation in Bacillus Calmette-Guérin Treatment

The study by Pietzak et al. provided genomic information and clinical findings related to Bacillus Calmette-Guérin (BCG) treatment [22]. We studied the association between *KDM6A* mutation and the recurrence rate after BCG treatment. There was no significant difference in response between the males and females undergoing BCG treatment (Table 5).

### 3.7. Gender-Biased Prognostic Role of KDM6A Mutation in Cisplatin-Based Chemotherapy

The study by Taber et al. gave us multi-omics data and clinical outcomes related to cisplatin-based chemotherapy [25]. We studied the association between the *KDM6A* mutation and the clinical outcomes related to cisplatin-based chemotherapy. Among the patients receiving cisplatin-based chemotherapy, there was no significant difference between the males and females in either their response to this chemotherapy (Table 6) or in the Kaplan–Meier analysis of their prognosis (Figure 5A–C).

### 3.8. Gender-Biased Prognostic Role of KDM6A Mutation in Immune Therapy

Three studies (IMvigor 210 [26], the Samstein et al. study [19], and UC-GENOME [21]) offered genomic data and clinical findings related to immune therapy in UC. We examined the association between the *KDM6A* mutation and the clinical outcomes related to immune therapy in UC. The female patients with *KDM6A* mutation were significantly associated with poor outcomes (SD/PD) in the IMvigor 210 study (Table 7). Kaplan–Meier analysis showed that male patients with *KDM6A* mutation were associated with a favorable prognosis in the study by Samstein et al., whereas no significant difference in prognosis was observed in the female patients (Figure 6A–C). Kaplan–Meier analysis showed that female patients with *KDM6A* mutation were associated with a poor prognosis in the UC-GENOME study but that no significant difference in prognosis was observed in the male patients in that study (Figure 6D–F and Appendix A). These results indicate that among patients undergoing immune therapy, female patients with *KDM6A* mutation were associated with a poorer prognosis than that of male patients.

### 3.9. Gender-Biased Role of KDM6A Mutation in Immune Phenotypes and Tumor Immune Microenvironment

The IMvigor 210 and UC-GENOME studies reported findings related to immune phenotypes. We analyzed the findings of gender-biased associations between the immune phenotypes. The rate of immune-inflamed type was higher in males than in females in the patients with *KDM6A* mutation in the IMvigor 210 and UC-GENOME studies (Table 8). A recent study reported the immune signatures by single-sample GSEA analysis in the IMvigor 210 study [28], whereas the study from UC-GENOME reported on the immune signatures by CIBERSORT analysis [21]. This allowed us to analyze gender-biased associations between the immune signatures in relation to *KDM6A* mutation status in these two cohorts. The CD8+ fraction, APC co-inhibition fraction, and type-1 IFN fraction were significantly lower in females than in males in the patients with *KDM6A* mutation from the IMvigor 210 study (Figure 7A–C). Additionally, the UC-GENOME study showed that the Ayers T-cell inflamed fraction, Ayers IFN fraction, and Bindea T-cell fraction were significantly lower in the females than in the males with *KDM6A* mutation (Figure 7D–F). In contrast, there was no significant gender-biased difference in these immune fractions in patients with *KDM6A* wild-type (Figure 7A–F). These results suggest that the cold immune microenvironment was observed more frequently in females than in males among the patients with *KDM6A* mutation.

## 4. Discussion

*KDM6A* was identified as one of the loss-of-function mutations of six X-chromosome genes, and it occurred more frequently in males than females [12]. However, in this study, the *KDM6A* mutation rate was significantly higher in females than males in BLCA and UTUC in the AACR GENIE cohort, the UROMOL study, and the UTUC study by Fujii et al. A recent study reported that the *KDM6A* mutation rate was higher in females than males with NMIBC [31]. Our findings showed that the *KDM6A* mutation rate was higher in females than males with NMIBC in the UROMOL study and the study by Clinton et al., but not in that by Pietzak et al. We could not obtain detailed clinicopathological information for the AACR GENIE and ICGC cohorts. What is more, most of the patients in the TCGA cohort had MIBC. Although the *KDM6A* mutation rate may vary based on factors such as NMIBC distribution, cohort size, demographic variations, sample quality, and other clinical variables, the larger sample size of approximately 4000 in the AACR GENIE cohort might suggest that *KDM6A* mutations are more prevalent in females than in males. A recent study has shown that urothelium-specific knockout of *KDM6A* increased BLCA risk in female but not male mice [32]. Collectively, *KDM6A* mutation may contribute to the biological gender differences seen in UC.

The present study showed that female patients with *KDM6A* mutation were associated with poor prognosis in three cohorts related to immune therapy. Indeed, immune fractions such as CD8+ and IFN were lower in females than males with *KDM6A* mutation. In other cohorts related to BCG treatment and cisplatin-based chemotherapy, no significant difference in prognosis was found between the females and males with *KDM6A* mutation. These results indicate that there appears to be a gender-biased immune response in patients with *KDM6A* mutation and that female patients with *KDM6A* mutation had tumors with cold immune status. However, the mechanism for this is not known. As we mentioned above, *KDM6A* is crucial in regulating gene expression through modifying histones. Loss of function mutation in the *KDM6A* gene leads to dysregulated gene expression [29]. A recent study found that *KDM6A* mutation downregulated specific signaling pathways that are involved in the immune system and attenuated the tumor immune response [34]. It is thought that the UTY gene on the Y chromosome may compensate for the mutation of *KDM6A* on the X chromosome [35,36]. What is more, the loss of the Y chromosome drives growth by evasion of tumor immunity [37]. These findings suggest that the Y chromosome might be involved in the gender-biased immune response in patients with *KDM6A* mutation.

The present study found that the *KDM6A* mutation rate was about 20–30%. *KDM6A* is one of the genes covered by the targeted tumor-sequencing test, such as MSK-IMPACT [38], indicating that KDM6A mutation may be easily analyzed. These results suggest that *KDM6A* may be a promising treatment target as well as a biomarker. A mutation in *KDM6A* activates the *EZH2* function [39]. Recent studies have shown that an *EZH2* inhibitor is effective against *KDM6A* mutation [40,41]. Tazemetostat is an EZH2 inhibitor approved for epithelioid sarcoma and follicular lymphoma with an *EZH2* mutation by the US FDA [42]. However, to date, no clinical trials have reported the testing of EZH2 inhibitors in BLCA. A recent study showed that an EZH2 inhibitor enhanced antigen presentation and overcame resistance to anti-PD-1 therapy in head and neck cancer [43]. The present study showed that there was a gender-biased difference in the immune environment in patients with *KDM6A* mutation, and male patients with *KDM6A* mutation had tumors with immune hot status. Collectively, these findings suggest that combination treatment with an EZH2 inhibitor and immune therapy may be promising for male patients with *KDM6A* mutation.

There were several limitations in this study. We considered all mutation types as mutated types for mutation analysis because not all databases provided detailed mutation information. In the future, we will analyze the impact of detailed mutation types of *KDM6A* on immune response. In this study, we downloaded the processed data from several cohorts. The clinicopathological distribution of these cohorts was different. In the future, we plan to download the raw data from these cohorts and validate our current findings after normalizing the data. Although the present study showed no significant gender-biased difference in prognosis in the BCG treatment and chemotherapy cohorts, the sample size of these cohorts was relatively small. A cohort with a larger sample size will be necessary to verify the present data. Although we showed a gender-biased difference in *KDM6A* mutation in immune response using the public databases, the mechanism was not clarified. Our findings need to be validated by in vitro and in vivo experiments. Our previous study showed that *KDM6A* was involved in cancer progression with a *TP53* mutation using conditional *KDM6A* knockout mice [44]. In the future, we will analyze the gender-biased effect of *KDM6A* mutation on immune response using conditional *KDM6A* knockout mice.

## 5. Conclusions

We found that the rate of *KDM6A* mutation was higher in females than in males in several cohorts. We also revealed that the female patients with *KDM6A* mutation were associated with a poorer prognosis than the male patients among the patients undergoing immune therapy. The female patients had an immune cold microenvironment compared to that of the male patients. These results indicate that the effect of *KDM6A* mutation on immune therapy varied according to gender, and the status of *KDM6A* may be a promising biomarker in immune therapy.

## Figures and Tables

**Figure 1 cancers-17-00356-f001:**
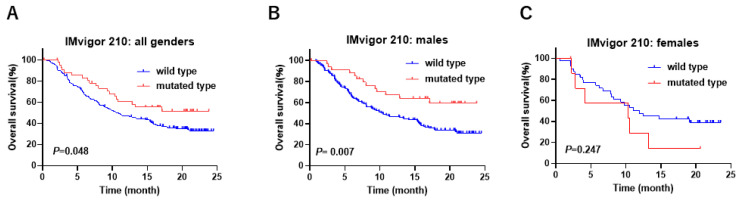
The gender-biased difference in overall survival according to *KDM6A* mutation status in the IMvigor 210 study. (**A**) In all genders, (**B**) in males, and (**C**) in females.

**Figure 2 cancers-17-00356-f002:**
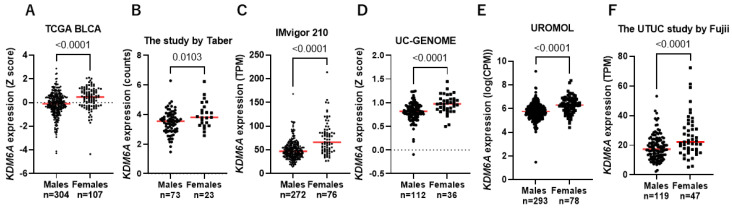
The expression of *KDM6A* in males and females. (**A**) In TCGA, (**B**) in the study by Taber et al. [25], (**C**) in the IMvigor 210 study, (**D**) in UC-GENOME, (**E**) in UROMOL, and (**F**) in the study by Fujii et al. [27]. The red line represents the mean bar.

**Figure 3 cancers-17-00356-f003:**
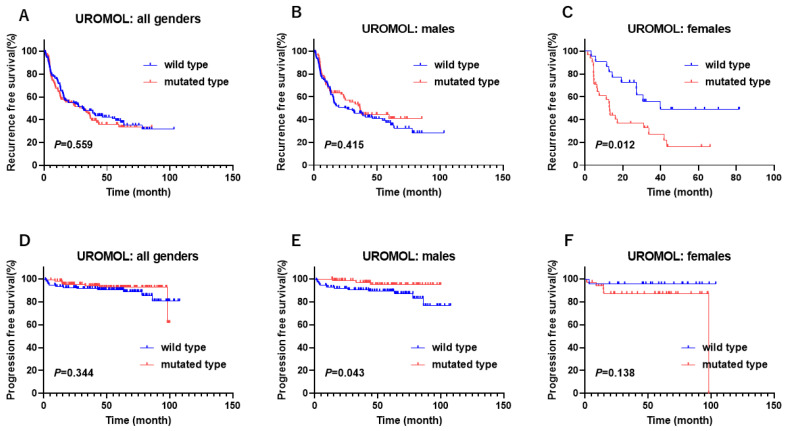
The gender-biased difference in survival according to *KDM6A* mutation status in non-muscle invasive bladder cancer. (**A**) In all genders, (**B**) in males, and (**C**) in females in recurrence-free survival from the UROMOL study. (**D**) In all genders, (**E**) in males, and (**F**) in females in progression-free survival from the UROMOL study.

**Figure 4 cancers-17-00356-f004:**
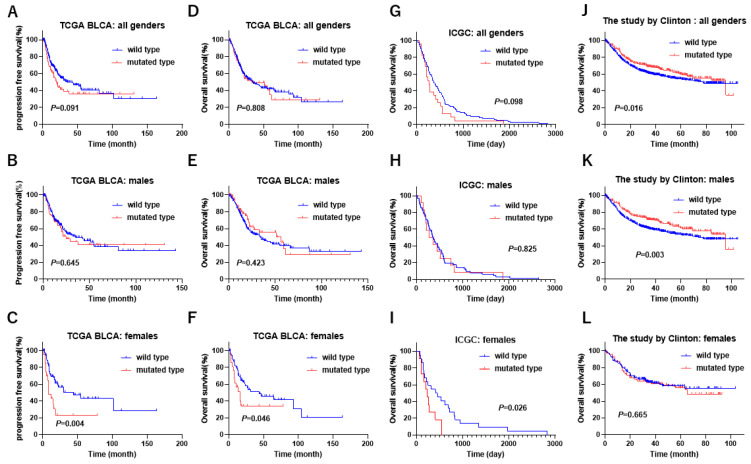
The gender-biased difference in survival according to *KDM6A* mutation status. (**A**) In all genders, (**B**) in males, and (**C**) in females in progression-free survival from the TCGA BLCA dataset. (**D**) In all genders, (**E**) in males, and (**F**) in females in overall survival from the TCGA BLCA dataset. (**G**) In all genders, (**H**) in males, and (**I**) in females in overall survival from the ICGC dataset. (**J**) In all genders, (**K**) in males, and (**L**) in females in overall survival from the study by Clinton et al. [20].

**Figure 5 cancers-17-00356-f005:**
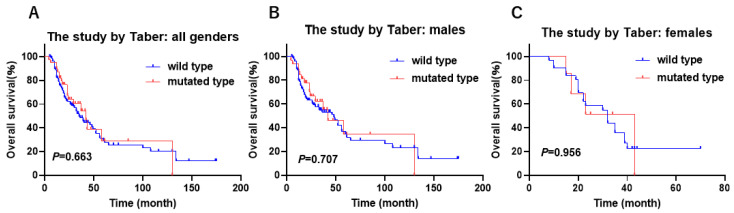
The gender-biased difference in overall survival according to *KDM6A* mutation status in the study by Taber et al. [25]. (**A**) In all genders, (**B**) in males, and (**C**) in females.

**Figure 6 cancers-17-00356-f006:**
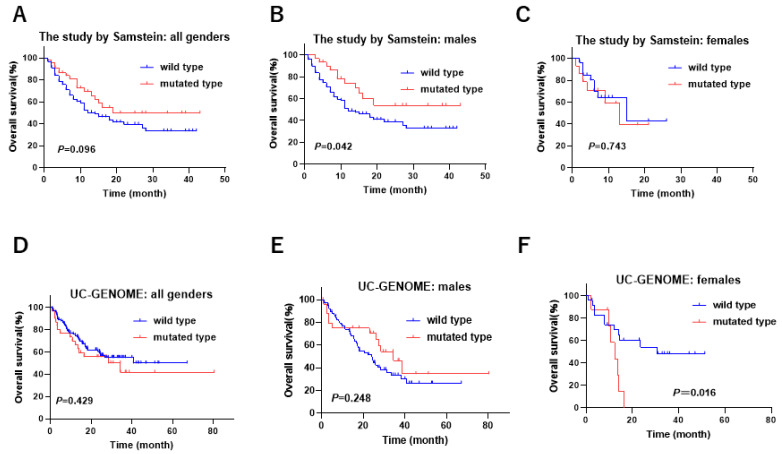
The gender-biased difference in overall survival according to *KDM6A* mutation status in the immune therapy cohorts. (**A**) In all genders, (**B**) in males, (**C**) and in females in the study by Samstein et al. [19]. (**D**) In all genders, (**E**) in males, and (**F**) in females in the UC-GENOME.

**Figure 7 cancers-17-00356-f007:**
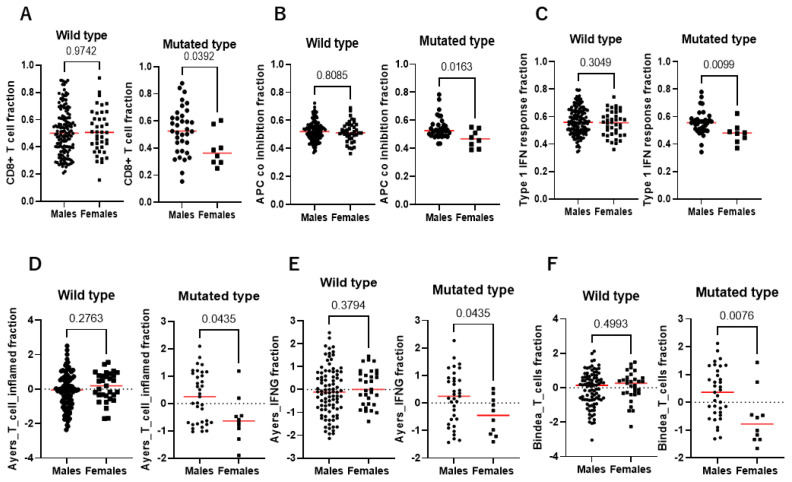
The gender-biased difference in immune fraction according to *KDM6A* mutation status in the immune therapy cohorts. (**A**) CD8+ T cell fractions, (**B**) APC co-inhibition fraction, and (**C**) type 1 IFN response fraction in the IMvigor 210 study. (**D**) Ayers_T_cell_inflamed fraction, (**E**) Ayers_IFNG fraction, and (**F**) Bindea_T_cells fraction in UC-GENOME. The red line represents the mean bar.

**Table 1 cancers-17-00356-t001:** The gender difference in KDM6A mutation rate in non-muscle invasive bladder cancer.

	*KDM6A* Status		*p*-Value ^a^
	Wild Type	Mutated Type	
UROMOL study (n = 435)			
Males (n = 344)	218 (63%)	126 (37%)	<0.001
Females (n = 91)	40 (44%)	51 (56%)	
The study by Pietzak [22] (n = 105)			
Males (n = 80)	44 (55%)	36 (45%)	0.336
Females (n = 25)	11 (44%)	14 (56%)	
The study by Clinton [20] (n = 66)			
Males (n = 46)	31 (67%)	15 (33%)	0.004
Females (n = 20)	6 (30%)	14 (70%)	

^a^ *p*-values were calculated with Fisher’s exact test.

**Table 2 cancers-17-00356-t002:** The gender difference in KDM6A mutation rate in bladder cancer.

	*KDM6A* Status		*p*-Value ^a^
	Wild Type	Mutated Type	
AACR GENIE cohort (n = 4008)			
Males (n = 3065)	2248 (74%)	817 (26%)	0.014
Females (n = 943)	653 (69%)	290 (31%)	
TCGA (n = 410)			
Males (n = 303)	224 (74%)	79 (26%)	0.562
Females (n = 107)	76 (71%)	31 (29%)	
ICGC (n = 514)			
Males (n = 396)	307 (78%)	89 (22%)	0.302
Females (n = 118)	86 (73%)	32 (27%)	
UC-GENOME (n = 168)			
Males (n = 128)	95 (74%)	33 (26%)	0.921
Females (n = 40)	30 (75%)	10 (25%)	
The study by Taber [25] (n = 165)			
Males (n = 127)	95 (75%)	32 (25%)	0.378
Female s(n = 38)	31 (81%)	7 (19%)	

^a^ *p*-values were calculated with Fisher’s exact test.

**Table 3 cancers-17-00356-t003:** The gender difference in KDM6A mutation rate in upper tract urothelial carcinoma.

	*KDM6A* Status		*p*-Value ^a^
	Wild Type	Mutated Type	
AACR GENIE cohort (n = 731)			
Males (n = 456)	372 (82%)	84 (18%)	0.019
Females (n = 275)	204 (74%)	71 (26%)	
The study by Fujii [27] (n = 235)			
Males (n = 162)	132 (81%)	30 (19%)	0.031
Females (n = 73)	50 (68%)	23 (31%)	

^a^ *p*-values were calculated with Fisher’s exact test.

**Table 4 cancers-17-00356-t004:** The gender-biased difference in the prognosis of *KDM6A* mutation (The AACR GENIE cohort) in bladder cancer.

	*KDM6A* Status		*p*-Value ^a^
	Wild Type	Mutated Type	
All gender (n = 3921)			
Alive (n = 2319)	1615 (70%)	704 (30%)	<0.001
Death (n = 1602)	1206 (75%)	396 (25%)	
Males (n = 3006)			
Alive (n = 1781)	1250 (70%)	531 (30%)	<0.001
Death (n = 1225)	942 (77%)	283 (23%)	
Females (n = 915)			
Alive (n = 538)	365 (68%)	173 (32%)	0.482
Death (n = 377)	264 (70%)	113 (30%)	

^a^ *p*-values were calculated with Fisher’s exact test.

**Table 5 cancers-17-00356-t005:** The gender-biased difference in recurrence of *KDM6A* mutation in BCG treatment (The study by Pietzak [22]).

	*KDM6A* Status		*p*-Value ^a^
	Wild Type	Mutated Type	
All gender (n = 69)			
Recurrence (n = 33)	18 (55%)	15 (45%)	0.401
No recurrence (n = 36)	16 (44%)	20 (56%)	
Males (n = 55)			
Recurrence (n = 27)	16 (59%)	11 (41%)	0.222
No recurrence (n = 28)	12 (43%)	16 (57%)	
Females (n = 14)			
Recurrence (n = 6)	2 (33%)	4 (67%)	0.531
No recurrence (n = 8)	4 (50%)	4 (50%)	

^a^ *p*-values were calculated with Fisher’s exact test.

**Table 6 cancers-17-00356-t006:** The gender-biased difference in response based on *KDM6A* status in cisplatin-based chemotherapy (The study by Taber [25]).

	*KDM6A* Status		*p*-Value ^a^
	Wild Type	Mutated Type	
All gender (n = 164)			
CR/PR (n = 104)	82 (79%)	22 (21%)	0.423
SD/PD (n = 60)	44 (73%)	16 (27%)	
Males (n = 126)			
CR/PR (n = 78)	61 (78%)	17 (22%)	0.354
SD/PD (n = 48)	34 (71%)	14 (29%)	
Females (n = 38)			
CR/PR (n = 26)	21 (81%)	5 (19%)	0.848
SD/PD (n = 12)	10 (83%)	2 (17%)	

^a^ *p*-values were calculated with Fisher’s exact test. CR: complete response, PR: partial response, SD: stable disease, and PD: progression disease.

**Table 7 cancers-17-00356-t007:** The gender-biased difference in response based on *KDM6A* status in bladder carcinoma treated with immune therapy.

	*KDM6A* Status		*p*-Value ^a^
	Wild Type	Mutated Type	
UC-GENOME			
All gender (n = 104)			
CR/PR (n = 41)	30 (77%)	11 (23%)	0.256
SD/PD (n = 63)	52 (83%)	11 (17%)	
Males (n = 75)			
CR/PR (n = 30)	23 (77%)	7 (23%)	0.266
SD/PD (n = 45)	39 (86%)	6 (13%)	
Females (n = 19)			
CR/PR (n = 5)	4 (80%)	1 (20%)	0.703
SD/PD (n = 14)	10 (71%)	4 (29%)	
IMvigor 210			
All gender (n = 227)			
CR/PR (n = 60)	44 (73%)	16 (27%)	0.083
SD/PD (n = 167)	140 (83%)	27 (16%)	
Males (n = 179)			
CR/PR (n = 49)	33 (67%)	16 (33%)	0.008
SD/PD (n = 130)	111 (85%)	19 (15%)	
Females (n = 48)			
CR/PR (n = 11)	11 (100%)	0 (0%)	0.032
SD/PD (n = 37)	29 (78%)	8 (22%)	

^a^ *p*-values were calculated with Fisher’s exact test. CR: complete response, PR: partial response, SD: stable disease, and PD: progression disease.

**Table 8 cancers-17-00356-t008:** The gender difference in KDM6A mutation in immune phenotypes in IMvigor 210 and UC-GENOME in urothelial carcinoma.

	*KDM6A* Status		*p*-Value ^a^
	Wild Type	Mutated Type	
IMvigor 210 (n = 230)			
Males (n = 184)			
Desert (n = 48)	40 (83%)	8 (17%)	0.361
Excluded (n = 84)	69 (82%)	15 (18%)	
Inflamed (n = 52)	38 (73%)	14 (27%)	
Females (n = 46)			
Desert (n = 13)	9 (69%)	4 (31%)	0.296
Excluded (n = 21)	19 (90%)	2 (10%)	
Inflamed (n = 12)	10 (83%)	2 (17%)	
UC-GENOME (n = 151)			
Males (n = 112)			
Desert (n = 1)	1 (100%)	0 (0%)	0.721
Excluded (n = 68)	50 (74%)	18 (26%)	
Inflamed (n = 43)	31 (72%)	12 (28%)	
Females (n = 39)			
Desert (n = 0)	0 (0%)	0 (0%)	0.228
Excluded (n = 21)	14 (67%)	7 (33%)	
Inflamed (n = 18)	15 (83%)	3 (17%)	

^a^ *p*-values were calculated with Fisher’s exact test.

## Data Availability

All data generated or analyzed during this study are included in this article or the Appendix A.

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
