# Peer review of "The Gender-Biased Differential Effect of KDM6A Mutation on Immune Therapy in Urothelial Carcinoma: A Public Database Study"

_cancers, 2025, doi:10.3390/cancers17030356_

Round 1
Reviewer 1 Report
Comments and Suggestions for Authors
The author investigated the correlation between KDM6A mutation and effectiveness of immune therapy using public database set. As a result, female patients with KDM6A mutation had poorer prognosis and revealed the immune cold condition compared with patients with KDM6A wild type. The mechanisms of gender differences in bladder cancer progression have been unsolved and the theme of this study is very informative.
For publication in this journal, the author should clarify the questions as below.
1. As author described in limitation, they showed that there was a gender-biased difference in immune response, the mechanism was unclear. In addition, it is unclear whether KDM6A mutation was the cause or the effect. The author should clarify these points.
2. The author described that KDM6A might apply as biomarker in immune therapy, however, it is not realistic to perform widespread genomic testing. The author should discuss in detail what kind of operation is possible.
Author Response
Thank you for reviewing our manuscript. We did not take into account that IMvigor 210 is composed of bladder cancer and upper tract urothelial carcinoma. Additionally, we added data in non-muscle invasive bladder cancer (UROMOL study). Therefore, we slightly changed the structure of our manuscript. We have carefully revised the manuscript based on the reviewer's comments.
Comment 1: As author described in limitation, they showed that there was a gender-biased difference in immune response, the mechanism was unclear. In addition, it is unclear whether KDM6A mutation was the cause or the effect. The author should clarify these points.
Response 1: Thank you for your valuable comment. Basically, KDM6A plays a crucial role in regulating gene expression through modifying histones. Loss of function mutation in KDM6A leads to dysregulated gene expression. Therefore, we speculate that KDM6A mutation may cause a gender-biased difference in immune response. However, the mechanism was unclear. We added a detailed explanation of the function of KDM6A.
These statements were described in the revised results and discussion (page 4, line 140-142 and page14, line 314-316).
Comment 2: The author described that KDM6A might apply as biomarker in immune therapy, however, it is not realistic to perform widespread genomic testing. The author should discuss in detail what kind of operation is possible.
Response 2: Thank you for your valuable comment. KDM6A is one of the genes covered by targeted tumor-sequencing tests, such as MSK-IMPACT. KDM6A mutation is relatively easy to analyze by genomic testing.
These statements were described in the revised discussion (page 14, line 324-326).

Reviewer 2 Report
Comments and Suggestions for Authors
SeKino et al. investigate the expression and mutation of KDM6A, an X chromosome inactivation (XCI) gene, using publicly available databases related to immune therapy, chemotherapy, and BCG treatment. Their analysis demonstrates that KDM6A expression is higher in females than males in several urothelial cancer cohorts, and they highlight a gender-biased difference in immune cell fractions based on KDM6A mutation status in immune therapy cohorts.
The manuscript has potential for publication in this journal if the following points are addressed to improve rigor and clarity:
1. Provide a detailed classification of the KDM6A mutations analyzed. Are all included mutations loss-of-function mutations, or are other mutation types considered? Clarifying this point would enhance the interpretability of the findings.
2. Not all cohorts showed a gender difference in KDM6A mutation rates. The authors should discuss possible reasons for this inconsistency, such as cohort size, demographic variations, or methodological differences. A deeper exploration of these factors would provide greater context for the results.
3. When comparing IMvigor 210, the Samstein et al. study, and UC-GENOME data, only the IMvigor 210 dataset identified female patients with KDM6A mutations as having a poorer prognosis compared to male patients. The authors should explore and explain why this observation was specific to IMvigor 210, considering potential biological or technical reasons.
4. The manuscript would benefit from greater detail in the methods section: How were sequencing results normalized across different datasets? What methods were used for immune signature analysis? Were statistical analyses adjusted for potential confounding factors like age or tumor stage?
5. Figure 2 Discrepancy: There appears to be a mismatch in Figure 2D between the number of points on the plot and the "n=30" noted below. The authors should review the original data and ensure the accuracy of the figures.
6. The authors should position their findings within the broader field and clearly outline the study's limitations, particularly the reliance on public datasets and the lack of experimental validation. Highlighting these limitations transparently could strengthen the manuscript's credibility.
Comments on the Quality of English Language
The manuscript's readability would improve by eliminating repetition and overly verbose sentences.
Author Response
Thank you for reviewing our manuscript. We did not take into account that IMvigor 210 is composed of bladder cancer and upper tract urothelial carcinoma. Additionally, we added data in non-muscle invasive bladder cancer (UROMOL study). Therefore, we slightly changed the structure of our manuscript. We have carefully revised the manuscript based on the reviewer's comments.
Comment 1: Provide a detailed classification of the KDM6A mutations analyzed. Are all included mutations loss-of-function mutations, or are other mutation types considered? Clarifying this point would enhance the interpretability of the findings.
Response 1: Thank you for your critical comments. In this study, we considered all mutation types as “mutated type” because not all databases provided detailed mutation information. Additionally, it may not be clear which specific mutation in the KDM6A gene holds biological significance. In the future, we will analyze the role of the detailed mutation type of KDM6A in immune therapy. We added the description to the method and limitation sections.
These statements were described in the revised method and limitation (page 3, line 111-113 and page14, line 338-341).
Comment 2: Not all cohorts showed a gender difference in KDM6A mutation rates. The authors should discuss possible reasons for this inconsistency, such as cohort size, demographic variations, or methodological differences. A deeper exploration of these factors would provide greater context for the results.
Response 2: We totally agree with this comment and revised the description in the discussion.
These statements were described in the revised results (page13, line 299-300).
Comment 3: When comparing IMvigor 210, the Samstein et al. study, and UC-GENOME data, only the IMvigor 210 dataset identified female patients with KDM6A mutations as having a poorer prognosis compared to male patients. The authors should explore and explain why this observation was specific to IMvigor 210, considering potential biological or technical reasons.
Response 3: Thank you for your comment. Did the reviewer imply the UC-GENOME cohort?
Only the UC-GENOME dataset showed that female patients with KDM6A mutations had a poorer prognosis compared to male patients. Supplementary figure 2 and 3 showed that females were associated with poor prognosis compared to males in the patients with KDM6A mutation in both IMvigor 210 and UC-GENOME cohorts. The ratio of the mutated type of KDM6A was slightly different between IMvigor 210 and UC-GENOME. We speculate that this difference in mutation rate may result in the difference in prognosis.
Comment 4: The manuscript would benefit from greater detail in the methods section: How were sequencing results normalized across different datasets? What methods were used for immune signature analysis? Were statistical analyses adjusted for potential confounding factors like age or tumor stage?
Response 4: Thank you for your comment. In this study, we did not combine the different cohorts to analyze the role of KDM6A mutation. We downloaded the processed data, which had already been normalized in these cohorts. Regarding immune signature analysis, we also downloaded the processed immune fraction data (ssGSEA and cybersort). We totally agree with your concerns about confounding factors. In the future, we plan to download the raw data from several cohorts. We added the description to the material and method section and discussion section.
These statements were described in the revised material and method (page 3, line 97-119).
Comment 5: Figure 2 Discrepancy: There appears to be a mismatch in Figure 2D between the number of points on the plot and the "n=30" noted below. The authors should review the original data and ensure the accuracy of the figures.
Response 5: Thank you for pointing out our mistake. We revised the figure 2D.
Comment 6: The authors should position their findings within the broader field and clearly outline the study's limitations, particularly the reliance on public datasets and the lack of experimental validation. Highlighting these limitations transparently could strengthen the manuscript's credibility.
Response 6: We totally agree with your opinion. Our findings need to be validated by the in vitro and in vivo experiments. We added the descriptions in the limitation section.
These statements were described in the revised discussion (page 15, line 349-350).

Reviewer 3 Report
Comments and Suggestions for Authors
Dear Authors, an interesting study on the differential effect of KDM6A in urothelial carcinoma patients stratified based on gender. Prior to my recommendation, please answer or consider the following:
(1) Consider abbreviating “bladder cancer” as “BLCA” instead of “BCa” because the latter is sometimes used to describe breast cancer.
(2) Italicize gene symbols throughout the manuscript. Most of the time, symbols are not italicized (I found only a few examples where they were italicized, e.g., in line 112).
(3) Please rewrite your objective/aim of the study in the Abstract (A simple summary is ok). Currently, in the Abstract you wrote what was identified (i.e., mainly KDM6A); however, the aim of the study is very well written in the Introduction (lines 77-80), i.e., you intended to analyze the prognostic role of six genes related to X-chromosome.
(4) Please double-check if in-text citations should have round or square brackets according to the publisher’s guidelines.
(5) Introduction, line 49: consider changing “T1 tumors are an aggressive subtype” to “T1 tumors constitute aggressive subtype”.
(6) Materials and Methods: mention what kind of data were downloaded from each repository/cohort. For example, based on the TCGA example, did you download counts or processed data? If counts, were they according to updated STAR protocol? If processed, were they TPM, RPKM, FPKM, or other? In general, the methodology lacks details. There is no information about the acquisition of immune fraction data. Results include Kaplan-Meier plots but no description of generating such plots is provided. The same applies to jitter plots. Unless all results were visualized based on GraphPad and/or JMS17? If yes, then I still think that at least a sentence or two should be added.
(7) Results, line 105: consider changing “Escape Genes from X Chromosome Inactivation” to “Genes Escaping from X Chromosome Inactivation”.
(8) Figure 2: add units on Y-axes (TPM/RPKM/FPKM/other?) and remember to mention about log2 if it was applied. Moreover, if possible, please use some kind of color for the median so that it can be easily visible for all subfigures.
(9) Table 1: move “Bladder cancer” and “UTUC” or separate them in the first column so that it can be easily visible that UTUC is in fact based on AACR only. Moreover, it seems that only AACR shows any significant results for BLCA and UTUC. Could you discuss this in the text? A sentence or two would be enough. As for Table 1, I suggest changing all “N.S.” to actual p-values even if they are not significant. This way it would be much more evident that the results did not meet significance but some were close to it. You can also bold statistically significant p-values or add some sort of significance marks (“*”) next to them.
(10) I presume that your survival analysis is limited to overall survival only, based on the fact that it might be the only endpoint available in some cohorts. However, please consider adding other endpoints such as DFS/RFS/PFS for cohorts for which they are available, e.g., for TCGA. The mentioned endpoints offer earlier presentation since events due to disease recurrence/progression occur earlier than death from the disease.
(11) Results, line 168: remove excessive period marks at the end of the line. Same in line 193 but in the middle.
(12) Results, line 172: capitalize “bacillus”.
(13) Figure S2C: add “s” in “females”.
(14) Tables 2-7: I suggest to change all “N.S.” to actual p-values even if they are not significant. This way it would be much more evident that results did not meet significance but some were close to it. You can also bold statistically significant p-values or add some sort of significance marks (“*”) next to them.
(15) I see that your immune fraction analysis (mostly Figure 6) is based on IMvigor 210 and UC-GENOME. However, it is possible to enrich this analysis with TCGA data because immune estimations can be calculated for this cohort using, e.g., TIMER 2.0. Please consider doing so.
(16) Discussion, lines 275-277: is the part about Erdafitinib necessary? I think it does not gravitate toward the remaining part of a paragraph, at least it is worse than in the case of, e.g., Tazemetostat which is an EZH2 inhibitor, whereas EZH2 was associated with KDM6A mutational status.
(17) Supplementary Materials, lines 306-310: either describe all subfigures here or delete them and instead put appropriate annotations so that subfigure symbols would not be required to understand the content of supplementary figures. Moreover, in Figure S1 title, consider changing “5 genes” to actual gene symbols, or alternatively put them in brackets within the sentence.
(18) Disclaimer/Publisher’s note is not present after References.
Author Response
Thank you for reviewing our manuscript. We did not take into account that IMvigor 210 is composed of bladder cancer and upper tract urothelial carcinoma. Additionally, we added data in non-muscle invasive bladder cancer (UROMOL study). Therefore, we slightly changed the structure of our manuscript. We have carefully revised the manuscript based on the reviewer's comments.
Comment 1: Consider abbreviating “bladder cancer” as “BLCA” instead of “BCa” because the latter is sometimes used to describe breast cancer.
Response 1: Thank you for your valuable suggestion. We changed BCa to BLCA through the manuscript.
Comment 2: Italicize gene symbols throughout the manuscript. Most of the time, symbols are not italicized (I found only a few examples where they were italicized, e.g., in line 112).
Response 2: Thank you for pointing out our mistakes. We italicize gene symbols throughout the manuscript.
Comment 3: Please rewrite your objective/aim of the study in the Abstract (A simple summary is ok). Currently, in the Abstract you wrote what was identified (i.e., mainly KDM6A); however, the aim of the study is very well written in the Introduction (lines 77-80), i.e., you intended to analyze the prognostic role of six genes related to X-chromosome.
Response 3: Thank you for your valuable suggestion. We revised the abstract.
These statements were described in the revised abstract (page 1, line 30-33).
Comments 4: Please double-check if in-text citations should have round or square brackets according to the publisher’s guidelines.
Response 4: Thank you for your valuable suggestion. We revised the reference style throughout the manuscript.
Comments 5: Introduction, line 49: consider changing “T1 tumors are an aggressive subtype” to “T1 tumors constitute aggressive subtype”.
Response 5: Thank you for your valuable suggestion. We revised the description according to the reviewer’s comment.
These statements were described in the revised introduction (page 1, line 62).
Comments 6: Materials and Methods: mention what kind of data were downloaded from each repository/cohort. For example, based on the TCGA example, did you download counts or processed data? If counts, were they according to updated STAR protocol? If processed, were they TPM, RPKM, FPKM, or other? In general, the methodology lacks details. There is no information about the acquisition of immune fraction data. Results include Kaplan-Meier plots but no description of generating such plots is provided. The same applies to jitter plots. Unless all results were visualized based on GraphPad and/or JMS17? If yes, then I still think that at least a sentence or two should be added.
Response 6: Thank you for your essential comment.
#1 We downloaded the processed data from these cohorts. These cohorts provided the data in different units (Z-score, TPM, read counts). We revised figure 2A-F.
#2 Regarding immune fraction data, we downloaded the processed immune fraction data (ssGSEA and cybersort) and added the detailed information to the Materials and Methods section.
#3 Kaplan-Meier plots and jitter plots were generated by GraphPad.
These statements were described in the revised Materials and Methods (page 3, line 115-119 and page 4, line 128-129).
Comment 7: Results, line 105: consider changing “Escape Genes from X Chromosome Inactivation” to “Genes Escaping from X Chromosome Inactivation”.
Response 7: Thank you for your comment. We revised the title of the result according to the reviewer’s comment.
Comment 8: Figure 2: add units on Y-axes (TPM/RPKM/FPKM/other?) and remember to mention about log2 if it was applied. Moreover, if possible, please use some kind of color for the median so that it can be easily visible for all subfigures.
Response 8: Thank you for your valuable suggestion. We revised Figure 2A-E.
Comment 9: Table 1: move “Bladder cancer” and “UTUC” or separate them in the first column so that it can be easily visible that UTUC is in fact based on AACR only. Moreover, it seems that only AACR shows any significant results for BLCA and UTUC. Could you discuss this in the text? A sentence or two would be enough. As for Table 1, I suggest changing all “N.S.” to actual p-values even if they are not significant. This way it would be much more evident that the results did not meet significance but some were close to it. You can also bold statistically significant p-values or add some sort of significance marks (“*”) next to them.
Response 9: Thank you for your valuable suggestion. We separated Table 1 into new Table 1(Bladder cancer) and new Table 2(UTUC). We added the new findings from the UROMOL study and the study by Fujii. The rate of KDM6A mutation was higher in females than in males in these cohorts. We discussed the limitation of our findings related to the KDM6A mutation rate in the first paragraph of the discussion. We revised the Table 1-8.
These statements were described in the revised discussion (page 13, line 299-300).
Comment 10: I presume that your survival analysis is limited to overall survival only, based on the fact that it might be the only endpoint available in some cohorts. However, please consider adding other endpoints such as DFS/RFS/PFS for cohorts for which they are available, e.g., for TCGA. The mentioned endpoints offer earlier presentation since events due to disease recurrence/progression occur earlier than death from the disease.
Response 10: Thank you for your valuable suggestion. We analyzed the gender-biased prognostic role of KDM6A mutation at PFS and OS from TCGA cohorts and revised Figure 4. We added the data in NMIBC from the UROMOL study and added the RFS and PFS data in Figure 3.
These statements were described in the revised Results (page 6, line 179-184).
Comment 11: Results, line 168: remove excessive period marks at the end of the line. Same in line 193 but in the middle.
Response 11: Thank you for pointing out our mistake. We removed the excessive period marks.
Comment 12: Results, line 172: capitalize “bacillus”.
Response 12: Thank you for pointing out our mistake. We revised the description.
Comment 13: Figure S2C: add “s” in “females”.
Response 13: Thank you for pointing out our mistake. We revised Figure S2C.
Comments 14: Tables 2-7: I suggest to change all “N.S.” to actual p-values even if they are not significant. This way it would be much more evident that results did not meet significance but some were close to it. You can also bold statistically significant p-values or add some sort of significance marks (“*”) next to them.
Response 14: Thank you for pointing out our mistake. We revised Tables 1-8.
Comment 15: I see that your immune fraction analysis (mostly Figure 6) is based on IMvigor 210 and UC-GENOME. However, it is possible to enrich this analysis with TCGA data because immune estimations can be calculated for this cohort using, e.g., TIMER 2.0. Please consider doing so.
Response 15: We totally agree with your suggestions. We analyzed the immune estimations in TCGA cohort and found that there was no significant difference between males and females in patients with KDM6A mutation.
Comment 16: Discussion, lines 275-277: is the part about Erdafitinib necessary? I think it does not gravitate toward the remaining part of a paragraph, at least it is worse than in the case of, e.g., Tazemetostat which is an EZH2 inhibitor, whereas EZH2 was associated with KDM6A mutational status.
Response 16: Thank you for your critical suggestions. We removed the description related to Erdafitinib.
Comment 17: Supplementary Materials, lines 306-310: either describe all subfigures here or delete them and instead put appropriate annotations so that subfigure symbols would not be required to understand the content of supplementary figures. Moreover, in Figure S1 title, consider changing “5 genes” to actual gene symbols, or alternatively put them in brackets within the sentence.
Response 17: Thank you for valuable suggestion. We deleted the subfigure symbols and revised the description in Supplementary Materials section.
These statements were described in the revised Supplementary Materials (page 15, line 364-371).
Comment 18: Disclaimer/Publisher’s note is not present after References.
Response 18: Thank you for your valuable comment. We added Disclaimer/Publisher’s note after References.
These statements were described in the revised manuscript (page 17, line 474-477).

Round 2
Reviewer 1 Report
Comments and Suggestions for Authors
none
Reviewer 2 Report
Comments and Suggestions for Authors
The author has addressed my questions, and the quality of the data and figures in the manuscript has improved. I have no further questions.
Reviewer 3 Report
Comments and Suggestions for Authors
Thank you for addressing my comments, I endorse the publication.